# Multi-Extraction and Quality of Protein and Carrageenan from Commercial Spinosum (*Eucheuma denticulatum*)

**DOI:** 10.3390/foods9081072

**Published:** 2020-08-06

**Authors:** Alireza Naseri, Charlotte Jacobsen, Jimmy J. P. Sejberg, Tommy Ewi Pedersen, Jan Larsen, Karin Meyer Hansen, Susan L. Holdt

**Affiliations:** 1Research Group for Bioactives—Analysis and Application, The National Food Institute, Technical University of Denmark (DTU Food), Kemitorvet, Building 204, DK-2800 Kgs. Lyngby, Denmark; alireza@food.ku.dk (A.N.); suho@food.dtu.dk (S.L.H.); 2Innovation, CP Kelco, Ved Banen 16, DK-4623 Lille Skensved, Denmark; jimmy.sejberg@cpkelco.com (J.J.P.S); tommy.ewi.pedersen@gmail.com (T.E.P.); jan.larsen@cpkelco.com (J.L.); KarinMeyer.hansen@cpkelco.com (K.M.H.)

**Keywords:** algae, industrial seaweeds, vegan protein, combined extraction, bioactive compounds, sustainability, bioeconomy, functionality

## Abstract

Seaweeds contain many valuable compounds that can be used in the food industry. Carrageenan is a polysaccharide which has been extracted from seaweed for centuries and is used as a texturizer in food and non-food products. However, seaweeds contain compounds other than carrageenan, such as proteins, which could also be extracted. This extraction should be done without compromising the industrial scale carrageenan extraction yield and quality. This study aimed at up-stream protein extraction from red seaweed *Eucheuma denticulatum* by using of an optimized enzyme-assisted extraction, including of an aqueous/enzymatic treatment followed by alkaline extraction, and then the commercial carrageenan extraction. The protein extraction efficiency of four enzymes was evaluated including Celluclast^®^ 1.5L, Shearzyme^®^ 500 L, Alcalase^®^ 2.4 L FG and Viscozyme^®^ L at a concentration of 0.0, 0.1, 0.2 and 0.4% (*w*/*w*). To avoid detrimental effects on carrageenan, all the experiments were performed at pH 7 at room temperature. The results showed that 0.2% *w*/*w* Alcalase^®^ or Viscozyme^®^ added individually achieved the highest protein extraction efficiencies (59 and 48%, respectively) at pH 7 and room temperature (*p* < 0.05). Determination of the most common carrageenan quality parameters indicated that using any of these enzymes had no negative effect on the carrageenan yield and quality.

## 1. Introduction

Seaweeds contain many industrial ingredients, but also other compounds of interest, including bioactive compounds, which can be used in the food industry. Carrageenan is a polysaccharide, which has been extracted from seaweed for centuries and is used as a texturizer in food and non-food products. *Eucheuma denticulatum,* with the commercial name spinosum, is one of the main red seaweeds used in carrageenan production, and this seaweed constituted almost 20% of the carrageenan-containing seaweeds harvested worldwide in 2015 [1]. Methods of extracting carrageenan from seaweed are well known, and such methods involve alkaline extraction of seaweed at high temperatures (generally around 100–120 °C) [2]. However, the commercially used alkaline extraction of seaweed usually results in a carrageenan product, which also contains other compounds, such as proteins, antioxidants, and pigments [3].

It would be advantageous to have a method of preparing carrageenan, where other compounds including bioactive compounds from seaweeds could be extracted as value-added products, without decreasing the carrageenan yield and the functional quality of the carrageenan.

Furthermore, the global demand for protein is increasing and is expected to escalate further in the coming decades—mostly due to population growth, which must be matched by increased food production [4]. There is a good reason to develop new technologies for the industrial extraction of vegan proteins from seaweeds, since some species of seaweed have an interesting amino acid profile close to that of animal protein. For example, the ratio between essential amino acids (EAA) and total amino acids (AA) in *Palmaria palmata* was 44–53% while this score for soya and egg protein was 39 and 47% respectively [5]. The global carrageenan production in 2014 amounted to 60,000 tonnes, with a value of US$ 626 million in 2014. From these numbers, it can be estimated that the total dried seaweed consumption for this production, was at least 300,000 tonnes per year. The protein content of these types of seaweed are in the range of 4–28% [6]. If just half of the total amount of protein could be extracted, more than 20,000 tonnes of a new high-value protein product would be obtained per year. [6,7] The variation in protein content in seaweeds used in carrageenan production could be attributed to differences in seasonality, growth conditions in the environment, or source and species of resource [6,8].

Although *E. denticulatum* with 4–5% protein content has the lowest level of protein in comparison with other seaweeds such as *Palmaria palmata*, *Furcellaria lumbricalis* or *Chondrus crispus,* it should be considered that the global industrial utilization of this seaweed for carrageenan production was 45000 dry tonnes in 2015 which is equal to 20% of total harvest. Therefore, large amounts of industrial seaweed are available [1,6]. Hence, an improved method of extracting carrageenan from seaweed would be advantageous, and in particular, a method enabling the extraction of proteins and other bioactive compounds in addition to carrageenan, without any detrimental effects on carrageenan yield and quality.

In recent years, varieties of processes for the extraction of protein from various seaweed species have been reported. These processes include enzyme-assisted extraction (EAE) [7,9,10], physical processes [11,12], chemical extraction [13,14], as well as novel techniques such as ultrasound-assisted extraction (UAE) [15,16], pulsed electric field (PEF) extraction [12,17], and microwave-assisted extraction (MAE) [4,18].

Some of these studies evaluated the effect of enzymatic treatment, alkaline pretreatment, and process conditions on viscosity and gel strength of carrageenan; extracted from different red seaweeds commonly used in carrageenan production. As an example, Azevedo et al. (2015) studied the effect of pre-extraction alkali treatment on the chemical structure and gelling properties of extracted hybrid carrageenan from *C. crispus* and *Ahnfeltiopsis devoniensis* [19]. The results showed that increasing the KOH content, and the pre-treatment time improved the gelling properties in both seaweeds [19]. In addition, the effect of process conditions on the viscosity and gel strength of semi-refined carrageenan (SRC), produced from red seaweed *Kappaphycus alvarezii* was studied by Anisuzzaman et al. in 2014. The experimental results showed that gel viscosity increased with the decrease of cooking time, cooking temperature and potassium hydroxide (KOH) concentration (% *w*/*w*). In contrast, gel strength increased when cooking time, cooking temperature and KOH concentration (% *w*/*w*) increased. [20]. Moreover, in another study, the development of a high yielding carrageenan extraction method from *Eucheuma cottonii* using cellulase and the fungi *Aspergillus niger* was investigated [21]. However, no other studies have dealt with both protein and carrageenan extraction from *E. denticulatum,* as well as the effects of the protein extraction on the yield and quality of the extracted carrageenan.

The overall aim of this study was to design and develop a method to extract more than one product (multi-extraction), e.g., extraction of both protein and carrageenan, from *E. denticulatum* (spinosum). The extraction of protein from spinosum upstream the carrageenan extraction was optimized with regard to temperature, pH, the concentration of enzymes, and extraction time. The selected enzymes were in accordance with the previous study, which was successfully carried out on *P.palmata* [5]. The quality of the extracted protein was evaluated based on the amino acid profile, and the contents of essential amino acids (EAA). In addition, the effect of multi-extraction on the yield and quality of the isolated carrageenan (i.e., gel strength, turbidity, and viscosity) was evaluated.

## 2. Material and Methods

### 2.1. Seaweed Biomass and Preparation

CP Kelco (Lille Skensved, Denmark) provided the *Eucheuma denticulatum* seaweed used in their carrageenan production line. The seaweed biomass was harvested and shipped in semi-dried condition in big pack pallets from the Philippines and Indonesia to CP Kelco. The pallets were stored in a non-insulated warehouse at ambient outdoor temperature. The industrial samples were taken from different batches to obtain representative samples. All seaweed samples were dried at ambient temperature (as is the procedure in industrial-scale carrageenan production), after which the biomass was reduced to 0.5–1 cm particle size using a cutting mill (SM 2000, Retsch, Haan, Germany) in the lab of DTU Food. The milled seaweed was kept in plastic bags in a freezer at −20 °C until used. Dry matter (DM) content for all samples in this study was determined by drying the biomass in an oven at 105 °C until constant weight.

### 2.2. Enzymes and Chemicals

All enzymes, including a cellulase Celluclast^®^ 1.5 L, the xylanase Shearzyme^®^ 500 L, the protease (endo-peptidase) Alcalase^®^ 2.4 L FG, and the multi-enzyme mixture Viscozyme^®^ L which contains a wide range of carbohydrases, including arabinase, cellulase, β-glucanase, hemicellulase, and xylanase, were provided by Novozymes A/S (Bagsværd, Denmark). All chemicals used in this study were from Merck (St. Louis, IL, USA). All solvents used were HPLC grade and purchased from Lab-Scan (Dublin, Ireland). The standards for amino acids analysis were purchased from Sigma-Aldrich (St. Louis, MO, USA). HPLC grade water was prepared by a Milli-Q^®^ Advantage A10 water deionizing system from Millipore Corporation (Billerica, MA, USA).

### 2.3. Aqueous/Enzymatic Extraction

Semi-dried seaweed samples (25 g) were weighed in 1 L Erlenmeyer flasks. In accordance with the selected ratio in Table 1, deionized water was added to each of 1 L Erlenmeyer flasks. The sample was rehydrated for 1 h, and the enzymes were then added. Carrageenan releases from seaweed by increasing the temperature, and the enzymatic treatment was therefore performed at room temperature (20–22 °C), in order not to obtain simultaneous carrageenan and protein extraction. All treatments were run in triplicates (*n* = 3).

### 2.4. NAC-Assisted Alkaline Extraction

The conditions for the alkaline extraction were modified from those reported by Harnedy and FitzGerald [22] and were done in accordance with the study by Naseri et al. for the extraction of protein from *P. palmata* [5]. In summary, after the enzymatic treatment (see Section 2.3), the alkaline extraction was don and repeated three times with a solution containing 1 g/L of NAC and 4 g/L of NaOH. A laboratory orbital shaker was used at a speed of 120–130 rpm and ambient temperature for 1.5 h. (Figure 1). The liquid fractions recovered were pooled together with the liquid fraction recovered from the enzymatic extraction. The pooled liquid fraction was stored at 4 °C overnight before precipitation of protein by the addition of acid (HCl).

### 2.5. Post-Extraction Solid Residue (PESR)

At the end of the extraction process, the filter-cakes mentioned above were placed in plastic containers. In order to avoid negative effects of high temperature on protein quality, all filter-cakes were dried in an oven at 40–45 °C. Then, the samples were ground to powder and kept in the freezer at −20 °C prior to further analysis.

### 2.6. Experimental Plan

The proposed multi-extraction of proteins and carrageenan from spinosum seaweed is shown in Figure 1. This includes enzymatic treatment, NAC (see explanation above) assisted extraction and protein separation prior to the normal single product carrageenan production/extraction. In order to study and optimize this multi-extraction, the experimental setup consisted of five steps. The first step was to determine the best conditions for temperature, pH, the ratio of seaweed to water and enzyme combination. In accordance with the results reported in the literature [5,23], the ratios of seaweed to water selected were 1:15, 1:20 and 1:25. Enzyme concentrations of 0.0, 0.1, 0.2 and 0.4% *w*/*w* were used. All the enzyme treatments were carried out in duplicates for 4, 6 or 8 h at pH 7 to avoid detrimental effect on carrageenan quality.

In the second step, a variety of enzymes were evaluated at different concentrations followed by N-acetyl-L-cysteine (NAC) assisted extraction. NAC, L-cysteine-hydrochloride monohydrate and β-mercaptoethanol are reducing agents significantly increase the yield of alkaline soluble nitrogen [22]. Therefore, the combination of NaOH and NAC as a food-grade reducing agent was selected to increase the extraction of proteins in the current study. In this step, different enzymes were used at the selected optimal condition (Table 1), after which the effect of protein extraction on the yield of carrageenan extraction (step 3) and carrageenan quality (step 4) was evaluated. In the last step, the amino acid profiles for the selected treatments were compared. All treatments were run in triplicates (*n* = 3).

### 2.7. Protein Precipitation and Centrifugation

Protein precipitation was performed by lowering the pH of the solution to the isoelectric point of the protein (3.5). The pH of the pooled protein extract was lowered to 3.5 by the addition of aqueous hydrochloric acid (2 M). Subsequently, the mixture was centrifuged at 4400 g for 15 min at 4 °C and then the supernatant and solid residue were separated and stored in the freezer at −20 °C. The solid fractions were freeze-dried and milled prior to the analysis (*cf.* above).

### 2.8. Total Nitrogen, Protein Content and Extraction Efficiency

Total nitrogen content was measured by a DUMAS nitrogen/protein analyzer using a fully automated rapid MAX-N exceed (Elementar Analysensysteme GmbH, Langenselbold, Germany). For this purpose, 150–250 mg of dried fractions and 3–4 mL of liquid fractions were fed into the analyser. Protein content was calculated by multiplying the nitrogen content by a conversion factor of 5 [24]. All treatments were run in triplicates (*n* = 3).

The protein extraction efficiency for every treatment was calculated based on the below equation [5]:(1)$$\mathrm{Extraction}\;\mathrm{efficiency}\;\%\;=\frac{\mathrm{Protein}\;\mathrm{content}\;\mathrm{before}\;\mathrm{extraction}-\;\mathrm{Protein}\;\mathrm{content}\;\mathrm{after}\;\mathrm{extraction}}{\mathrm{Protein}\;\mathrm{content}\;\mathrm{before}\;\mathrm{extraction}}\times 100$$

To do the calculation faster, the below equation was used:(2)Extraction efficiency (%)=100% −Protein recovered in PESR % 

### 2.9. Carrageenan Extraction and Yield

After protein extraction, the seaweed samples were freeze-dried, milled and passed through a 1 mm mesh sieve to obtain uniform particle size. The carrageenan was extracted using a water-extraction technique. In brief, the seaweeds were soaked in milli-Q water overnight (5% *w*/*v*). The pH of the suspension was adjusted (pH 7.5–8.5) by KOH and the carrageenan was extracted at 99 °C for 1.5 h in a water bath with shaking. The carrageenan was separated from the seaweed residue by filtration using filter aid and subsequently isolated by precipitation in isopropanol at the ratio of 1:3 (seaweed/alcohol). The samples were freeze-dried, and yields were determined by weighing [25]. All treatments were run in triplicates (*n* = 3).

### 2.10. Carrageenan Gel Strength

The gelling properties of the isolated carrageenan was evaluated, as it would be performed in industry, in a dessert formulation (milk jelly) prepared from whole milk and sugar. The milk jelly ingredients were heated at 80 °C for 30 min, then filled directly into crystallization dishes. Finally, the gel was allowed to cool in a 5 °C water bath for 3–4 h and the top layer of the gel was then carefully removed using a wire cheese slicer. The gel and breaking strengths were measured with a Stable Micro System (SMS) Texture Analyser-TX.XT2 (Godalming, UK) with a plunger: 0.5 inch diameter; plunger speed: 1 mm/s; distance: 30 mm. The gel strength was measured at 4, 8 and 12 mm penetration of the gel. The breaking strength was measured at the first peak on the curve. The gel and breaking strengths were calculated as the averages of three measurements.

### 2.11. Turbidity of a 1.0% Solution of Carrageenan

The turbidity was determined using HACH 2100 N Turbidimeter. Turbidity was measured using a 1.0% (*w*/*w*) aqueous solution of carrageenan obtained in 2.9 which was prepared by the addition of 1.00 g carrageenan to 100 mL deionized water.

### 2.12. Food Chemicals Codex (FCC) Viscosity

The viscosity of the 1.5% aqueous solution of carrageenan was measured at 75 °C when making FCC-viscosity on a product standardized with sugar, and the amount of weighed material must be corrected to 1.5% pure material. FCC viscosity was measured at CP Kelco using a LVF viscometer (Brookfield, Middleboro, MA, USA.) fitted with a UL-adapter. The viscosity was measured at 30 rpm for 30 s.

### 2.13. Amino Acid Composition

The amino acid (AA) composition was determined as described by Farvin et al. (2010). To 50 mg dry weight of the sample was added 6 M aqueous hydrochloric acid, and the mixture was heated in the oven at 105 °C overnight. Following filtration through a 0.2 μm filter, derivatization was carried out using the EZ:Faast kit from Phenomenex A/S (Allerød, Denmark). The amino acid composition was determined using LC-MS (Agilent 1100 Series, LC/MSD Trap, Santa Clara, CA, USA) with an EZ:faast 4u AAA-MS new column (250 × 3.0 mm, Phenomenex) as described by Farvin et al. This procedure does, however, not allow for the detection of tryptophan (Trp) and cysteine (Cys) as both amino acids decompose during the acid hydrolysis [26].

### 2.14. Statistical Analysis

Statgraphics Centurion 18 (Statistical Graphics Corp., Rockville, MD, USA) was used for data analysis. Data were expressed as mean ± standard deviation, corresponding to three experimental replicates (*n* = 3). Firstly, by ANOVA test, multiple sample comparison analysis was performed to identify significant differences between samples. Secondly, mean values were compared using Duncan’s test. Differences between means were considered significant at *p* < 0.05.

## 3. Results and Discussion

### 3.1. The Optimization of Parameters for Enzymatic Treatment

In the present study, industrially utilized semi-dried *Eucheuma denticulatum* (spinosum) containing approximately 50% water and 3.8% DM protein was used as a raw material. Carrageenan is sensitive to low pH and will start to extract at high temperatures. Hence, to avoid detrimental effects on carrageenan quality, all experiments were performed at room temperature (˂30 °C) at pH ≥7. Based on preliminary experiments, a combination of Celluclast^®^ and Shearzyme^®^ was selected.

Table 1 shows the investigated process parameters, i.e., extraction time, seaweed to water ratio, enzyme concentration, and extraction efficiency. Generally, the obtained results show that the protein extraction efficiency after enzymatic extraction was significantly higher when using a 1:20 ratio of seaweed to water, an extraction time of 6 h, and enzyme concentrations of 0.2 or 0.4% *w*/*w*. Shorter extraction time also resulted in some degree of protein extraction, but with lower extraction efficiency. The treatment with the number of 11, 19, 20 and 24 had the highest extraction efficiencies. However, treatment number 19, which had the ratio 1:20 and was conducted for 6 h with 0.2% *w*/*w* of each enzyme, was selected as the optimal condition. This result was in accordance with the previous results reported by Naseri et al. in which applied enzyme-assisted extraction was the best among the tested to extract protein from red seaweed *Palmaria palmata* [5].

### 3.2. Different Enzymes for Protein Extraction

In general, seaweeds proteins are bound by other non-protein components such as polyphenols and polysaccharides within the cell. Furthermore, seaweeds proteins may be found within macro-molecular assemblies or cross-linked through disulphide bonds to polysaccharides [5,6,22,27,28]. Some previous studies have demonstrated that applying alkaline solutions for example sodium hydroxide (NaOH) significantly improves the solubility and extraction of highly water-insoluble proteins from seaweeds and microalgae [5,27,29]. In addition, food-grade NaOH is used in the food industry, and as an example, in the extraction of protein-rich ingredients from different plants for example mainly soybean and chickpea. Two other enzymes (Viscozymes^®^, and Alcalase^®^) were also tested and compared with the enzymes already tested in step 1 (Celluclast^®^, Shearzyme^®^). Furthermore, all these enzymatic extractions were followed by the NAC-assisted alkaline extraction containing 1 g/L of NAC and 4 g/L of NaOH to test for possible further optimization. In order to evaluate the efficiency of NAC-assisted alkaline extraction alone, the first treatment was done with no added enzymes resulting in a protein extraction efficiency of 15.7%. When Celluclast^®^ followed by NAC-assisted alkaline extraction was used, the results showed a steady increase from 19.4% at the concentration of 0.1% *w*/*w* enzyme to 37.9% for the concentration of 0.4% *w*/*w* enzyme. The results for Shearzyme^®^ were different, with the lowest efficiency of 12.3% at the concentration of 0.2% *w*/*w*, while the highest efficiency was 35.8% at the lower enzymatic concentration of 0.1% *w*/*w*. Moreover, it was noticeable that due to an unexplainable reason, the obtained results for protein extraction efficiency of Shearzyme^®^ at 0.2% *w*/*w* was lower than the treatment with no enzymes added.

The Viscozymes^®^ treatment resulted in a significant increase in extraction efficiency in comparison to Celluclast^®^ and Shearzyme^®^. The highest protein extraction efficiency was 48.5% at the concentration of 0.2% *w*/*w*. Viscozymes^®^ is a multi-enzyme complex containing a wide range of carbohydrases, including arabinose, cellulase, β-glucanase, hemicellulase, and xylanase, and this most likely explains why it was more efficient than the other carbohydrate degrading enzymes.

In the current study, the protease Alcalase^®^ was selected for testing in comparison with other non-protease enzymes, and the protein extraction efficiency of 59.4% was the significantly highest efficiency among all the experiments conducted in this study (concentration of 0.2% and tested at pH 7.0) (Table 2). Applying Alcalase^®^ led to the hydrolysis of peptide bonds that link amino acids together in the polypeptide chain forming the protein, and this is most likely the reason for the significantly improved protein extraction. It has also been reported that the protein extraction efficiency could be increased if there was no limitation and pre-defined framework with respect to pH and temperature [4,5,23]. For example, the extraction efficiency for *Palmaria palmata* could reach up to 90% by using the combination of Alcalase^®^ and Shearzyme^®^ or Alcalase^®^ and Celluclast^®^ at pH 8 and the concentration of 0.2% for each one at a temperature of 50 °C for 14 h [5]. However, as mentioned before, all experiments of this study were done at room temperature (<30 °C) and at neutral pH in order to the adverse effect on carrageenan quality and to avoid gelling during the extraction process.

### 3.3. Effect of the Protein Extraction Process on the Yield of Carrageenan Extraction

All the carrageenan extractions were performed based on an optimized method in the lab-scale done by Rhein-Knudsen et al. [25]. In the current study, the yield of isolated carrageenan extraction for the blank sample (with no enzymatic treatment and no NAC-assisted alkaline extraction) was 17.8%, while it increased to 23.8% when only NAC-assisted alkaline extraction with no enzymatic treatment was tested for protein extraction. Moreover, four different enzymatic treatments were selected to be compared with the blank samples, including Viscozymes (0.2% *w*/*w*) or Alcalase (0.2% *w*/*w*) or Celluclast (0.2% *w*/*w*) plus Shearzyme (0.2% *w*/*w*) at pH 6.0 and 7.0 followed by NAC-assisted alkaline extraction.

Table 3 shows that Viscozymes (0.2% *w*/*w*) or Alcalase (0.2% *w*/*w*) followed by NAC-assisted alkaline extraction had the lowest carrageenan isolation yield with 27.6 and 27.7%, respectively. Celluclast 0.2% *w*/*w* plus Shearzyme 0.2% *w*/*w* followed by NAC-assisted alkaline extraction at pH 6.0, and the same treatment at pH 7.0, had the significantly highest carrageenan yields. The results showed no significant difference between the two samples when treated by only Viscozymes (0.2% *w*/*w*) or Alcalase combined with NAC-assisted alkaline extraction (*p* < 0.05).

Varadarajan et al. compared the use of a cellulase from *Aspergillus niger*, and traditional boiling on the extraction of carrageenan from *Eucheuma cottonii* [9] They achieved the highest carrageenan yield when using the cellulase (45% by weight) compared to 37% for fungal treated and 37.5% for the traditional extraction methods. However, the viscosity of the cellulase-extracted carrageenan was lower than the one extracted by the traditional method. The decrease in viscosity could be explained by the presence of impurities bound to the carrageenan as the cellulase attacks the cell walls in the seaweeds, to release the carrageenan, and thus does not degrade the carrageenan structure itself [21].

### 3.4. Effect of the Protein Extraction Process on the Carrageenan Quality

Table 4 shows the results for gel strength, turbidity, intrinsic viscosity (IV) and Food Chemicals Codex (FCC) viscosity of 1.5% solution for the isolated carrageenan from the blank and different samples obtained after protein extraction.

The obtained results for gel strength indicated that in comparison with the blank sample, enzymatic treatment with Celluclast^®^ plus Shearzyme^®^ did not change gel strength in 4, 8 and 12 mm whereas the enzymatic treatment with single use of Alcalase^®^ or Viscozymes^®^ increased the gel strength significantly (Table 4). These treatments had the highest protein extraction efficiency; hence, the lowest amount of protein was left in the sample, and therefore, the carrageenan is most likely of higher purity.

The maximum gel strength (breaking strength) and breaking distance for the samples were significantly lower for the combination of Celluclast^®^ plus Shearzyme^®^ at pH 6 and 7 compared to the untreated sample. The same was the case for the sample treated with NAC only. In contrast, there was no significant difference between the samples treated by only Alcalase^®^ or Viscozymes^®^ and the blank sample (Table 4).

Regarding the turbidity, there was a considerable deviation in the obtained results. The highest turbidity was for the blank sample, but it was evident that all the treated samples, especially those treated by the combination of Celluclast^®^ plus Shearzyme^®^, had lower turbidity. The lowest turbidity was observed in the sample treated by Celluclast^®^ 0.2% *w*/*w* plus Shearzyme^®^ 0.2% *w*/*w* at pH 7 and followed by NAC-assisted alkaline extraction. Low turbidity is an advantage as it increases the possible applications of the carrageenan product. Therefore, protein extraction could be beneficial for expansion of carrageenan applications in different industries.

Intrinsic viscosity (IV) and FCC-viscosity as two parameters related to the gel viscosity indicated that there was no significant difference between the blank samples, and the samples treated by the combination of Celluclast^®^ plus Shearzyme^®^ or Viscozymes^®^ for protein extraction. There was also no statistical difference between the samples treated by Alcalase^®^, or by only NAC.

In summary, the obtained results in the present study indicate that Alcalase^®^ at 0.2% *w*/*w* and pH 7 or Viscozymes^®^ at 0.2% *w*/*w* and pH 7 are the optimal treatment to extract protein from spinosum. As mentioned before, the highest extraction efficiencies were for Alcalase^®^ at 0.2% *w*/*w* and pH 7, which led to a protein extraction efficiency of 59.4% and for Viscozymes^®^ at 0.2% *w*/*w* and pH 7, the extraction efficiency was 48.5%. It was obvious that in comparison with the Viscozymes^®^, the use of Alcalase^®^ will produce proteins/peptides with low molecular weight. Moreover, although the carrageenan yield for the combination of Celluclast^®^ plus Shearzyme^®^ was higher than when Viscozymes^®^ or Alcalase^®^ were used, the gel quality and in particular, the maximum gel strength (breaking strength) was lower for the combination of Celluclast^®^ plus Shearzyme^®^ compared to Viscozymes^®^ or Alcalase^®^. The results in Table 4 showed that by using Celluclast^®^ plus Shearzyme^®^, there was a negative effect on breaking strength compared to the blank sample. This was in compliance with the study performed by Varadarajan et al. (2009) [21].

Therefore, it could be concluded that using Alcalase^®^ at 0.2% *w*/*w* and pH 7 or Viscozymes^®^ at 0.2% *w*/*w* and pH 7 gave the higher protein extraction efficiency with no detrimental effects on carrageenan quality.

### 3.5. Comparison of Amino Acid Profiles

In the current study, the amino acid profile for the protein extracted by isoelectric precipitation at pH 3.5 for three different treatments were analyzed and compared, including Viscozymes^®^ 0.2% *w*/*w* or Alcalase^®^ 0.2% *w*/*w* or the combination of Celluclast^®^ 0.2% *w*/*w* plus Shearzyme^®^ 0.2% *w*/*w* which were followed by NAC-assisted alkaline extraction at pH 7. The reason for selecting these samples was in order to compare the sample with the highest carrageenan extraction yield with the samples with the highest protein extraction efficiencies.

The obtained results showed that protein extracted from the sample treated by the combination of Celluclast^®^ 0.2% *w*/*w* and Shearzyme^®^ 0.2% *w*/*w* had the highest amount of total amino acids with 50.8 mg/g DM. This was due to higher level of glutamic acid, aspartic acid, arginine and cysteine when compared to the protein extracted from the samples treated by Viscozymes^®^ or Alcalase^®^. The protein extracted from samples treated by Viscozymes^®^ 0.2% *w*/*w* or Alcalase^®^ 0.2% *w*/*w* had 36.2 and 25.5 mg/g DM, respectively (Table 5). The amino acids of highest amount were glutamic acid, aspartic acid, cystine, leucine, alanine, and valine. In addition, in the sample treated with Viscozymes^®^, the lysine content was higher than for the other treatments. However, it was noticeable that arginine was not detectable in the samples treated with Viscozymes^®^ or Alcalase^®^.

In comparison with the previous results obtained by the authors of this study with red seaweed *Palmaria palmata*, a protein extracted from *Euchema denticulatum* had a lower value of total amino acid (∑AA) and total essential amino acids (∑EAA) while the value for the ratio of EAA/AA was higher in protein extracted from *E. denticulatum*. The ratio was 27–42% for *P. palmata* while it was 46–54% for *E. denticulatum* [5]. The main reason could be due to the difference between the protein content of these two seaweeds.

The EAA/AA ratio of 46–54% of the protein of the *E. denticulatum* pellets in this experiment is in the same range as 48% EAA/AA in beef as a model organism for protein requirements, and the 47% EAA/AA in soybean [18,19]. The EEA/AA ratios calculated for beef and soybean were based on data for EEA and total protein content from Damodaran et al. [19]. In our calculation we have assumed that the protein content is equal to the sum of all amino acids. The branched chained amino acids (BCAA) are getting increasing attention, especially in fitness and body builder environments, since the leucine, isoleucine and valine are non-polar EAA that account for 35% of the EAA in the human muscles. The BCAA/AA ratio of the *E. denticulatum* pellets are 12–19% and therefore BCAA constitute a relative large part of the AA, which could make the product interesting for e.g., sport drinks and bars. However, it should be mentioned that protein content of beef and soybean is 18 and 40% respectively [19], compared to the 2.5–5.1% of the *E. denticulatum* pellets.

## 4. Conclusions and Future Perspectives

The present study successfully showed that it is possible to extract protein from *Eucheuma denticulatum,* in a multi-extraction setup, adding a new possible future protein resource with amino acid profiles comparable to meat (beef) and whey proteins, and further utilization going towards ‘no waste’ of our industrial resources. The best enzymes were Alcalase^®^ or Viscozymes^®^ at 0.2% *w*/*w*, and the maximum efficiency was increased by up to 60% of protein for Alcalase^®^. The present study furthermore demonstrated that the protein extraction process did not have a detrimental effect on the isolated carrageenan in the downstream carrageenan processing. For some parameters, the carrageenan behaved better than carrageenan extracted without the pre-extraction of protein. However, further investigation is needed to evaluate the bioactivity of both proteins and carrageenan before and after extraction.

## Figures and Tables

**Figure 1 foods-09-01072-f001:**
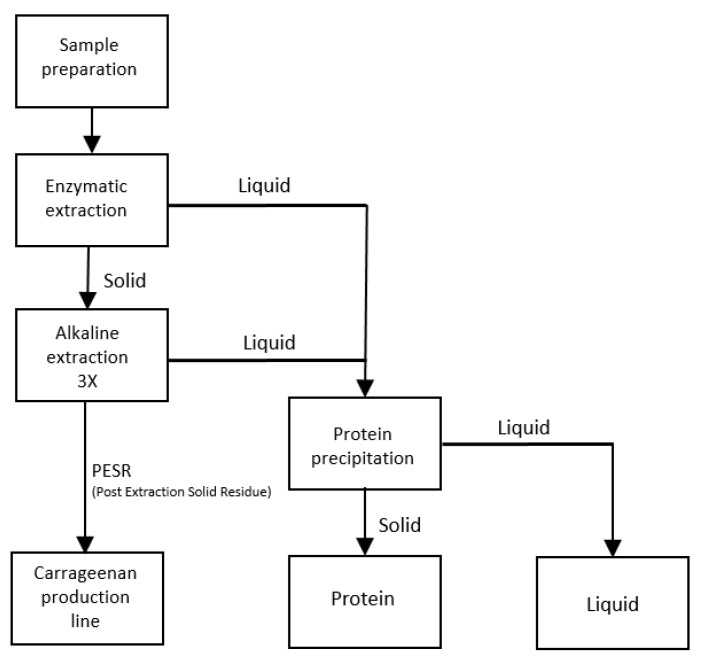
Process flow diagram for a proposed multi-extraction process of seaweed with both enzymatic extraction of proteins, followed by NAC assisted alkaline extraction before the actual industrial utilization of the spinosum seaweed for carrageenan production.

**Table 1 foods-09-01072-t001:** Protein extraction conditions (ratio, extraction time and enzymes concentration (on weight basis (*w*/*w*%)) and the average extraction efficiency (%) ± standard deviations of treated *Eucheuma denticulatum* seaweed. Different superscripts letters in the same column indicate significant difference at the confidence level of 95% (α = 0.05; *n* = 3).

Number	Ratio (Seaweed: Water)	Extraction Time (h)	The Concentration of Celluclast^®^ Plus Shearzyme^®^ (*w*/*w*%)	Extraction Efficiency (%)
1	1:15	4	0	7.2 ± 0.04 ^d^
2	1:15	4	0.1	10.9 ± 0.28 ^d^
3	1:15	4	0.2	9.2 ± 0.29 ^d^
4	1:15	4	0.4	12.7 ± 0.42 ^d^
5	1:15	6	0	8.1 ± 0.78 ^d^
6	1:15	6	0.1	9.0 ± 0.32 ^d^
7	1:15	6	0.2	12.3 ± 0.09 ^d^
8	1:15	6	0.4	11.9 ± 0.29 ^d^
9	1:15	8	0	8.8 ± 0.28 ^d^
10	1:15	8	0.1	9.4 ± 0.32 ^d^
11	1:15	8	0.2	14.5 ± 0.43 ^a,b,c^
12	1:15	8	0.4	13.7 ± 0.42 ^d^
13	1:20	4	0	7.4 ± 0.19 ^d^
14	1:20	4	0.1	7.3 ± 0.18 ^d^
15	1:20	4	0.2	9.9 ± 0.28 ^d^
16	1:20	4	0.4	12.7 ± 0.30 ^d^
17	1:20	6	0	10.1 ± 0.44 ^d^
18	1:20	6	0.1	11.5 ± 0.31 ^d^
19	1:20	6	0.2	15.1 ± 0.52 ^a,b^
20	1:20	6	0.4	15.5 ± 0.11 ^a^
21	1:20	8	0	8.8 ± 0.41 ^d^
22	1:20	8	0.1	11.6 ± 0.12 ^d^
23	1:20	8	0.2	12.9 ± 0.11 ^d^
24	1:20	8	0.4	14.2 ± 0.04 ^b,c^
25	1:25	4	0	8.9 ± 0.49 ^d^
26	1:25	4	0.1	8.9 ± 1.03 ^d^
27	1:25	4	0.2	12.9 ± 0.25 ^d^
28	1:25	4	0.4	11.8 ± 2.03 ^d^
29	1:25	6	0	9.3 ± 0.55 ^d^
30	1:25	6	0.1	9.8 ± 0.64 ^d^
31	1:25	6	0.2	10.7 ± 0.01 ^d^
32	1:25	6	0.4	12.9 ± 0.43 ^d^
33	1:25	8	0	9.4 ± 0.59 ^d^
34	1:25	8	0.1	10.7 ± 1.08 ^d^
35	1:25	8	0.2	12.1 ± 0.22 ^d^
36	1:25	8	0.4	12.6 ± 0.73 ^d^

Different letters indicate significant differences (*p* < 0.05).

**Table 2 foods-09-01072-t002:** Protein extraction efficiency (%) of the seaweed *Eucheuma denticulatum* treated with different enzymes followed by NAC-assisted alkaline extraction. Results are given as avearge ± standard deviation. Different superscripts letters in the same column indicate significant difference at the confidence level of 95% (α = 0.05; *n* = 3).

Samples	Efficiency %
Only enzymes (Celluclast + Shearzyme (0.2%) pH = 7)—No NAC-assisted alkaline extraction	15.1 ± 0.53 ^e,f^
No enzymes pH = 7	15.7 ± 0.12 ^e,f^
Celluclast (0.1%) pH = 7	19.4 ± 0.49 ^d,e,f^
Celluclast (0.2%) pH = 7	22.1 ± 8.89 ^d,e^
Celluclast (0.4%) pH = 7	37.8 ± 2.73 ^c^
Shearzyme (0.1%) pH = 7	35.8 ± 2.91 ^c^
Shearzyme (0.2%) pH = 7	12.3 ± 1.34 ^f^
Shearzyme (0.4%) pH = 7	24.8 ± 2.78 ^d^
Viscozymes (0.1%) pH = 7	36.0 ± 8.14 ^c^
Viscozymes (0.2%) pH = 7	48.5 ± 6.67 ^b^
Viscozymes (0.4%) pH = 7	41.8 ± 2.60 ^b,c^
Celluclast + Shearzyme (0.1%) pH = 7	20.5 ± 1.88 ^d,e,f^
Celluclast + Shearzyme (0.2%) pH = 7	39.1 ± 7.39 ^c^
Celluclast + Shearzyme (0.4%) pH = 7	24.8 ± 2.94 ^d^
Celluclast + Shearzyme (0.2%) pH = 6	17.7 ± 3.48 ^d,e,f^
Alcalase (0.2%) pH = 7	59.4 ± 1.41 ^a^

Different letters indicate significant differences (*p* < 0.05).

**Table 3 foods-09-01072-t003:** Effect of protein extraction on the yield (%) of isolated carrageenan of seaweed *Eucheuma denticulatum* treated with different enzymes followed by NAC-assisted alkaline extraction. Yields are given in average ± standard deviation. Different superscripts letters in the same column indicate significant difference at the confidence level of 95% (α = 0.05; *n* = 3).

Sample Description	Yield (%)
Blank (No enzymatic treatment, No NAC-assisted alkaline extraction)	17.8 ± 0.89 ^d^
No enzymatic treatment, only NAC-assisted alkaline extraction	23.8 ± 1.62 ^c^
Celluclast 0.2% + Shearzyme 0.2% (pH 6.0) + NAC-assisted alkaline extraction	31.9 ± 4.52 ^a,b^
Celluclast 0.2% + Shearzyme 0.2% (pH 7.0) + NAC-assisted alkaline extraction	35.5 ± 2.12 ^a^
Viscozymes 0.2% + NAC-assisted alkaline extraction	27.6 ± 0.97 ^b,c^
Alcalase 0.2% + NAC-assisted alkaline extraction	27.8 ± 3.26 ^b,c^

Different letters indicate significant differences (*p* < 0.05).

**Table 4 foods-09-01072-t004:** Effect of protein extraction of seaweed *Eurcheuma denticulatum* on the quality of carrageenan gel. Quality is given as averages ± standard deviations. Different superscript letters in the same column indicate significant difference at the confidence level of 95% (α = 0.05; *n* = 3).

Sample Treatments	Gel Strength 4 mm (g/cm)	Gel Strength 8 mm (g/cm)	Gel Strength 12 mm (g/cm)	Breaking Strength (g/cm)	Breaking Distance (mm)	Turbidity (NTU *)	IV	FCC ** Viscosity
Blank (No enzymatic treatment, No NAC-assisted alkaline extraction)	8.32 ± 0.13 ^b,c^	15.6 ± 0.29 ^b^	23.6 ± 0.50 ^b,c^	56.5 ± 3.53 ^a^	26.5 ± 0.22 ^a^	62.8 ± 25.2 ^a^	8.03 ± 0.22 ^a^	249 ± 24.7 ^a^
NO enzymatic treatment, Only NAC-assisted alkaline extraction, pH = 7	8.09 ± 0.01 ^c^	14.7 ± 0.14 ^c^	21.5 ± 0.57 ^c^	36.9 ± 6.71 ^b^	22.0 ± 2.33 ^c^	43.8 ± 11.2 ^a,b,c^	7.83 ± 0.51 ^a^	233 ± 33.9 ^a^
Celluclast 0.2% + Shearzyme 0.2% + NAC-assisted alkaline extraction, pH = 6	8.11 ± 0.42 ^c^	14.9 ± 0.61 ^c^	21.4 ± 0.01 ^c^	34.6 ± 1.62 ^b^	22.2 ± 0.92 ^c^	33.1 ± 12.0 ^b,c^	7.51 ± 0.31 ^a^	231 ± 49.5 ^a^
Celluclast 0.2% + Shearzyme 0.2% + NAC-assisted alkaline extraction, pH = 7	8.53 ± 0.01 ^b^	15.7 ± 0.14 ^b^	22.6 ± 0.43 ^c^	35.2 ± 1.50 ^b^	21.9 ± 0.11 ^c^	29.6 ± 5.72 ^c^	7.59 ± 0.24 ^a^	239 ± 24.0 ^a^
Viscozyme 0.2% + NAC-assisted alkaline extraction, pH = 7	11.6 ± 0.20 ^a^	19.1 ± 0.32 ^a^	25.4 ± 2.95 ^b^	59.0 ± 1.33 ^a^	24.6 ± 0.60 ^a,b^	48.1 ± 2.14 ^a,b,c^	7.89 ± 0.02 ^a^	222 ± 2.1 ^a^
Alcalase 0.2% + NAC-assisted alkaline extraction, pH = 7	11.7 ± 0.31 ^a^	19.4 ± 0.62 ^a^	28.3 ± 1.52 ^a^	59.1 ± 6.41 ^a^	24.5 ± 0.80 ^b^	54.2 ± 18.0 ^a,b^	7.48 ± 0.20 ^a^	218 ± 14.8 ^a^

NTU: Nephelometric Turbidity Unit, ** FCC: Food Chemicals Codex. Different letters indicate significant differences (*p* < 0.05).

**Table 5 foods-09-01072-t005:** Amino acid composition (mg amino acid/g DM), total essential amino acid content (∑ EAA), the essential amino acid ratio (EAA/AA) of the untreated samples of *Eucheuma denticulatum* seaweed, and extracted protein (pellet) obtained from the different selected enzymes used. The numbers are given as averages ± standard deviations of triplicates (n = 3).

Amino Acid	Untreated Seaweed	Viscozyme^®^ L	Alcalase^®^ 2.4 FL	Celluclast^®^ 1.5 L + Shearzyme^®^ 500 L
		Pellet	Pellet	Pellet
LYS ^(1)^	1.12 ± 0.04	2.26 ± 0.01	1.18 ± 0.20	1.99 ± 0.12
ALA	1.01 ± 0.08	2.29 ± 0.62	1.47 ± 0.02	3.00 ± 0.48
ARG	0.64 ± 0.01	n.d.	n.d.	4.74 ± 0.23
C-C ^(1)^	n.d	4.49 ± 1.03	6.00 ± 0.84	4.80 ± 0.06
MET ^(1)^	0.41 ± 0.03	1.00 ± 0.16	0.61 ± 0.05	1.27 ± 0.01
LEU ^(1)^	1.99 ± 0.27	2.64 ± 0.26	1.43 ± 0.01	3.35 ± 0.26
TYR ^(1)^	0.43 ± 0.09	1.18 ± 0.12	0.73 ± 0.05	1.31 ± 0.07
PHE ^(1)^	1.92 ± 0.07	2.08 ± 0.01	1.14 ± 0.10	2.59 ± 0.06
PRO	0.67 ± 0.05	1.48 ± 0.25	0.71 ± 0.04	1.93 ± 0.11
THR ^(1)^	2.37 ± 0.04	1.14 ± 0.21	0.56 ± 0.04	1.56 ± 0.42
ASP	2.62 ± 0.23	3.38 ± 1.04	2.44 ± 0.13	5.06 ± 0.93
SER	1.48 ± 0.01	0.99 ± 0.28	0.57 ± 0.06	1.49 ± 0.31
HYP	n.d.	n.d.	n.d.	n.d.
GLU	2.64 ± 0.19	6.09 ± 2.25	4.68 ± 0.19	7.68 ± 1.73
VAL ^(1)^	n.d	2.55 ± 0.26	1.10 ± 0.08	3.79 ± 0.89
HIS ^(1)^	0.25 ± 0.03	0.55 ± 0.00	0.33 ± 0.07	0.62 ± 0.01
ILE ^(1)^	1.08 ± 0.12	1.64 ± 0.03	0.57 ± 0.07	2.15 ± 0.18
GLY	0.89 ± 0.74	2.42 ± 0.59	1.97 ± 0.05	3.49 ± 0.42
∑ AA	19.5	36.2	25.5	50.8
∑ EAA	9.57	19.53	13.65	23.43
EAA/AA	0.49	0.54	0.54	0.46

^(1)^ Essential amino acids for human nutrition. n.d. means not detected.

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
