# Peer review of "Multi-Extraction and Quality of Protein and Carrageenan from Commercial Spinosum (Eucheuma denticulatum)"

_foods, 2020, doi:10.3390/foods9081072_

Round 1
Reviewer 1 Report
Authors here describe optimized multistep enzyme-assisted extraction of both proteins and carrageenan. This is a novel technique which has a potential to produce high value seaweed components with reduced bio-waste . However, author claims to isolate proteins and have shown amino acid composition to back these results. Were these proteins biologically active? Effect of treatment conditions on the activity of these proteins? Bioactivity of both proteins and carrageenan before and after extraction will strengthen the manuscript.
Here are my specific comments:
-Please edit typological error and grammatical mistakes in the manuscript.
-Several statements are missing their corresponding references, authors must use appropriate references, in introduction and discussion, upon siting literature. Please see few specific comments below.
Abstract and introduction (eg. line 27, 41, and 42) and so on, please use plural: seaweeds in the manuscript where applicable.
Double spacing between two words in abstract and introduction eg. line 30
Add reference to line 33. Alkaline extraction
Are there other methods for extraction of carrageenans? Commercial methods? If yes, please describe them in the introduction
Add reference to the statement in line 40
Line 42- Please include overlapping amino acids as compared to animal proteins.
Line 47-49: Missing link. Please add a line or two about variations in protein contents in seaweeds, then provide a reason for this variation.
Line 52: Add amounts of industrial E. denticulatum available after extraction, include reference.
Line 53-55: Same inference as line 36-38.
Line 57-60: Add reference for each technique.
Line 66- "both seaweeds" not "both seaweed"
Material and methods
Line 108: Was any optimization experiments performed to standardize the temperature for extraction process. Room temperature is not an accurate description and could be variable? What was the pH?
Line 124: ‘ground’ instead of grinded
Line 124: What is the significance of oven drying at 45 degrees?
Line 134: pH?
Results
Line 287: Effect of protein extraction process on carrageenan quality: add discussion with reference on gel strength?
Author Response
As you mentioned, this is a novel technique, which has a potential to produce high value seaweed components with reduced bio-waste. To the authors’ knowledge, there is no similar study and it makes it difficult to compare the obtained results. In order to keep the biological activity of the extracted protein, all the procedures were done at room temperature for avoiding any detrimental effects on protein quality. However, as we added to the manuscript, more study is needed to investigate about the effect of this novel technique on all the quality and biologically aspects of the products.
- The manuscripts was edited in terms of typological error and grammatical mistakes.
- The appropriate references were added to the manuscript.
- “Seaweeds” as a plural noun was corrected in the manuscript where applicable.
- Some values were added for the line 43-44 for amino acids as compared to animal proteins.
- The reason about variations in protein contents in seaweeds was already stated in lines 50-51, so we did not add additional information in line 47-49 as requested by the reviewer.
- The amount of available seaweed was added in lines 53 and 54.
- The references were added in line 60-62 for each technique.
- More explanations were added to elaborate why we use these enzymes at room temperature and neutral pH and why we use oven drying for the extracted protein at the specific temperature.
However, as I mentioned above, there is no literature available on some of the obtained results. Therefore, it is not possible to add any references and discussion because it is an industry standard.
For example for the below comment:
“Line 287: “Effect of protein extraction process on carrageenan quality: add discussion with reference on gel strength?”
Reviewer 2 Report
The manuscript deals with multi-extraction and quality of protein and carrageenan 3 from commercial spinosum (Eucheuma denticulatum). For this purpose, several experimental measurements were performed. The topic is interesting and falls within the Journal scope. The current version of the manuscript is well presented and organized, however some minor lacks should be improved. Please, follow the comments below:
Keywords: Personal note: Words included in the title should be avoided in this section, in order to expand the visibility of the manuscript.
Introduction: Recent articles needs to be cited, such as:
Torres, M.D., Flórez-Fernández, N., Domínguez, H. Integral utilization of red seaweed for bioactive production (2019) Marine Drugs, 17 (6), art. no. 314, .
Materials and methods: Replicates should be included in all sections.
Further justification of the used conditions or the corresponding reference should be included in this section.
Author Response
Thanks for the nice comments.
Reply to the comments from reviewer 2:
- The new keyword were added.
- The suggested reference was used and more reference were added.
Round 2
Reviewer 1 Report
All my comments have been addressed in Round 1. I do not have any further comments.